# Willingness to pay for an intervention that reduces soda consumption among a sample of middle-class adult Mexicans

**M. A. Colchero**[1]*, **Carlos M. Guerrero-López C. M.**[1], **Tonatiuh Barrientos-Gutiérrez T.**[2], **Jorge Salmerón J.**[3], **Sergio Bautista-Arredondo**[1]

1 Center for Health Systems Research, Instituto Nacional de Salud Pública, Cuernavaca, Morelos, Mexico,
2 Center for Population Health Research, Instituto Nacional de Salud Pública, Cuernavaca, Morelos, Mexico,
3 Centro de Investigación en Políticas, Población y Salud, Facultad de Medicina, Universidad Nacional Autónoma de México, Mexico City, Mexico

* acolchero@insp.mx

**Data Availability Statement:** The data set is available at: https://dataverse.harvard.edu/privateurl.xhtml?token=4388c024-47b3-4e8a-

## Abstract

### Background

Despite the growing public awareness of the adverse health effects of sugar-sweetened beverages (SSB) consumption in Mexico, little is known about the population's intention to reduce SSB consumption and the social value of interventions to accomplish such behavioral change. Therefore, the objective of this study was to assess the willingness to pay (WTP) for an intervention that reduces soda consumption by half in Mexico.

### Methods

We applied contingent valuation methods in a sample of 471 Mexican adults from a cohort study. We assessed the relative value of benefits by providing incremental information to participants in three scenarios: soda consumption reduction, + health benefits, + social benefits. To estimate factors associated with the WTP, we ran an interval regression.

### Results

87% of respondents reported they would like to reduce SSB consumption. High soda consumption, intention to reduce soda consumption and higher household income are associated with higher WTP. We found that the WTP increases as additional benefits are provided. The WTP, as a proportion of income, is higher for the lowest income level.

### Conclusion

The average WTP per person may be seen as the minimum amount the country should invest on interventions to reduce soda consumption.

ace3-73f442dc171e The DOI is: https://doi.org/10.7910/DVN/NY0JLA.

**Funding:** This work was funded primarily by Bloomberg Philanthropies. The study also received funding from the Instituto Nacional de Salud Pública and the National Institutes of Health R01DK108148. Beyond financial support, funders had no role in the study design, data col- lection, analyses, or interpretation.

**Competing interests:** No authors have competing interests.

## Introduction

Mexico faces a high burden of chronic diseases with a combined prevalence of overweight and obesity of 33.2% among children between 5 to 11 years old, 36.3% among adolescents, 72.5% among adults [1] and a prevalence of diabetes of 14.4% among adults [2]. One factor linked to weight gain, diabetes, and other chronic diseases is the consumption of sugar-sweetened beverages (SSB) [3–7]. Mexico is one of the highest SSB consumer in the world, estimated at 145 calories per capita sold per day in 2014 [8].

Several interventions have been implemented worldwide to reduce SSB consumption such as fiscal policies, regulation of food and beverages in schools, front-of-pack labeling, regulation of marketing directed to children, and subsidies to healthy food and beverages, among others. Current research is building a body of knowledge on the benefits and welfare implications of these interventions, including reductions in consumption [9], health benefits such as reductions in obesity, diabetes and other chronic disease and higher benefits for the poor [10, 11].

A recent fiscal policy implemented in Mexico reduced SSB household purchases by 7.5% two years after implementation of an excise tax of one peso per liter about a 10% increase in price [12]. The policy also increased sales of bottled water by 5% [13]. Even though modeling exercises predicted reductions in weight and diabetes [14], strokes, myocardial infarctions and ultimately mortality [15], the policy stirred up a political and scholarly debate, questioning the social value of a reduction in SSB consumption in Mexico. Specifically, despite the growing public awareness of the adverse health effects of SSB consumption, little is known about the population's intention to reduce SSB consumption and the social value of interventions to accomplish such behavioral change.

The primary objective of this study was to assess the intention to reduce soda consumption and the willingness to pay (WTP) for an intervention that reduces soda consumption by half in Mexico. We applied contingent valuation methods in a sample of Mexican adults who participate in a prospective cohort study, the Health Workers Cohort Study in Mexico [16]. We also assessed the relative value of individual and societal benefits by providing incremental information to participants and allowing them to change their WTP. Finally, we estimated the variation of WTP by sex, age, body mass index, intention to reduce soda consumption, consumption levels, and household income. The study is relevant given the high rates of SSB consumption and associated chronic disease in the country.

## Methods

The study protocol, questionnaires, procedures, and informed consent forms for the cohort study were approved by the corresponding IRBs of all participating institutions: the Mexican Social Security Institute (12CEI 09 006 14), the National Institute of Public Health (13CEI 17 007 36), and the Autonomous University of the Mexico State (1233008X0236). Signed written informed consent was obtained from all participants prior to enrollment.

As described below, we applied contingent valuation methods using a WTP questionnaire to a sample of individuals from a cohort study.

### The Health Workers Cohort Study

The Health Workers Cohort Study (HWCS) aims to study the association between genetics, lifestyle choices and cardiovascular health outcomes. Participants are IMSS employees and their families, living in the City of Cuernavaca in Morelos, Mexico. The cohort includes health providers, administrative staff, and academic personnel. The HWCS comprises three follow-up measurements between 2004 and 2017. In the baseline assessment conducted from March 2004 to April 2006, approximately 2,500 participants enrolled voluntarily. The second

assessment took place from 2010 to 2013. The third wave took place between 2016 and 2017. The size of the cohort by 2017 is 1,214 individuals. In addition to a survey on sociodemographic and behavioral characteristics, subjects completed anthropological and biological assessments. Further methodological details of the HWCS are reported elsewhere [16].

## Sample

We invited all cohort participants to answer the WTP questionnaire during the third wave through one of two approaches: 1) we reached and interviewed by telephone those who already had completed the third wave of instruments (26.5%); and 2) we interviewed participants in site during their biological assessments (73.5%). We excluded participants who reported no soda consumption in the last twelve months.

## WTP questionnaire

Contingent valuation methods are designed to elicit preferences and subjective valuation of hypothetical scenarios. Typically, respondents are asked to assess the economic value of an outcome through an iterative process designed to obtain their maximal WTP. We developed a contingent valuation questionnaire to elicit WTP among individuals who consumed soda in the previous 12 months, as reported in the third wave of the HWCS. Initially, we asked participants whether they wanted to reduce their consumption of soda. Then, we elicited their maximum WTP for a nutrition program designed to decrease in half their current soda intake as long as they remained in the program. We framed the WTP questions slightly different for participants according to their desire to cut soda consumption (see questionnaire in S1 Appendix).

The WTP section of our questionnaire included three questions meant to elicit the full economic value of the intervention described above, incrementally under three hypothetical scenarios. First, we measured the maximal WTP for the program without providing any additional information about its benefits, beyond reducing soda intake. The second scenario additionally offered a brief description of the potential health benefits associated with a reduction in soda consumption, before eliciting the maximum WTP. The third scenario added a social element of welfare, by exploring the WTP if the money collected by its sales were to be used to funding treatment of chronic diseases and providing potable water in the most deprived regions and public schools in Mexico.

Within each of the three scenarios, we elicited the maximum WTP through an iterative bidding process. For each scenario, we asked the participants whether they would be willing to pay X amount of money for the intervention. The initial amount X was randomly selected from a list of values between 50 and 500 Mexican pesos (either 50, 100, 200, 300, 400 or 500 pesos—roughly US $2.5, $5, $10, $15, $20 or $25, respectively). A bidding process followed. If participants responded "yes," we asked them whether they were willing to pay the next higher amount on the list—unless the random number was 500 pesos, in which case the process ended. If the participant responded "no" to the initial random amount, we asked for the immediate inferior amount—unless the random amount was 50 pesos. This iterative process was repeated up to three times for each scenario, and the final response represented the maximal WTP, expressed in an interval. S1 Fig provides a hypothetical example of all responses for a sequence of questions when the random start value is 100 pesos.

Data collection started in May 2017 in a subsample of the HWCS and ended in April 2018. Participants who attended their follow-up check-up were invited to participate in a face-to-face interview or by telephone.

## Statistical analysis

We first described the average WTP by intention to reduce soda consumption by scenario and the unadjusted distribution of the WTP for the three scenarios: 1) cut in soda consumption, 2) cut in soda consumption + explicitly accounting for individual health benefits, 3) adding social benefits. We also calculated a weighted average WTP by scenario taking the mid-point of each interval (lower and upper bounds) and assuming two values for the higher open-ended option of more than 500 pesos; 600 pesos and 1,000 pesos.

For empirical estimation, we ran an interval regression for each WTO scenario. Interval regressions are used when the dependent variable is expressed as a continuous variable or by intervals, with left- or right-censored data. The method is appropriate when the precise value of the dependent variable is unknown as there is interval censoring. In our study, the exact amount of the maximal WTP is unknown due to the right-censored maximal WTP at 500 pesos. Interval regressions estimations are based on the probability that a censored value fits within a range.

The independent variables included in the interval regression model were selected based on two criteria. First, WTP studies often adjust for sociodemographic characteristics because contingent valuation could vary by income, sex and age. Secondly, we included variables that could be associated with the WTP to reduce soda consumption such as intention to reduce soda consumption, soda consumption and overweight and obesity. Specifically, we included the following binary variables in the model: sex (1 for women, 0 otherwise); age group (1 for individuals 65 or more years old, 0 otherwise); intention to reduce soda consumption (1 = yes, 0 = no); high soda consumption (1 if soda expenditures were higher than the 75th percentile—30 pesos per week, 0 otherwise); overweight or obese (1 if body mass index is higher than 25, 0 otherwise); and household income tertiles. For household income, we imputed 16 missing observations using linear regression adjusted for sex and age, since missing values represent 3.4% of the sample size and this method allowed to preserve the structural relationships among the variables.

We used bootstrap with 1,000 replications and variance-covariance estimation to get bootstrap standard errors in the interval regression. Also, about 30% of participants belong to the same household so we included clustering at the household level in the interval regression.

Using the results of the interval regressions, we estimated the predicted values and upper and lower bounds of the maximal WTP for each scenario and by individual characteristics included in the model.

Finally, we calculated the ratio of the WTP relative to household income by dividing the predicted WTP for each scenario by the individually reported household income. We summarized the estimated ratios by income tertile.

We restricted the sample to respondents with no missing values in all of the variables listed above: we excluded nine observations corresponding to individuals with no BMI available, and 30 more due to inconsistent responses between the WTP scenarios. We encountered this problem at the beginning of data collection but stopped once we reinforced the training of interviewers. The analytic sample included 471 individuals from 510 participants in the WTP survey. We tested for differences in sex, high soda consumption, body mass index and age group between the analytical sample and the excluded observations using chi-square statistics. We also assessed whether the subsample of the WTP study was representative of the HWSC by testing for statistical differences in age, gender and education between our analytical sample and participants of the cohort in the third wave.

Statistical analyses were performed using Stata V13.1.

**Table 1. Characteristics of the 471 participants.**

| Variable | % or mean |
|---|---|
| Sex (female = 1) | 75% |
| Age group (more than 65 years old) | 23% |
| Education (university or higher) | 57% |
| Household monthly income (in pesos) by income group | |
| Low | 6,304.7 |
| Middle | 16,227.8 |
| High | 36,924.9 |
| High soda consumption (>$30 pesos/week) | 27% |
| Would like to reduce soda consumption | 87% |
| Overweight or obese (body mass index>25) | 66% |

## Results

Table 1 shows the characteristics of the analytical sample. From 471 participants, 75% were females, 23% were 65 years old and older and 57% completed a university degree. We found that 87% intended to reduce their soda consumption and 27% were high soda consumers. Among the 471 participants, 66% were overweight or obese.

Table 2 shows the average WTP by intention to reduce soda consumption and the distribution of the WTP intervals by scenario. Those who had the intention to reduce soda consumption had a higher WTP compared to those who had not intention to reduce soda consumption but their WTP. The average WTP for the soda consumption reduction scenario was 244.9 pesos for those with the intention to reduce soda consumption, 280.5 for the scenario that

**Table 2. Distribution of the sample population (n = 471) across willingness-to-pay ranges, by scenario.**

| Intention to reduce soda consumption (Mexican pesos) | Scenarios | | |
|---|---|---|---|
| | Soda consumption reduction | + health benefits | + social benefits |
| Intention to reduce soda consumption | 244.9 | 280.5 | 302.6 |
| No intention to reduce soda consumption | 162.5 | 212.0 | 264.5 |
| **WTP Interval (Mexican pesos)** | | | |
| Greater than 0 and lower than100 pesos | 29.1% [25.0, 33.2] | 26.1% [22.1, 30.1] | 21.0% [17.3, 24.7] |
| Between 100 and 200 pesos | 25.1% [21.1, 29.0] | 15.3% [12.0, 18.5] | 15.5% [12.2, 18.8] |
| Between 200 and 300 pesos | 9.6% [6.9, 12.2] | 15.4% [12.2, 18.9] | 13.0% [9.9, 16.0] |
| Between 300 and 400 pesos | 10.6% [14.1,8.1] | 9% [7.8, 13.4] | 12.1% [9.2, 15.0] |
| Between 400 and 500 pesos | 7.2% [4.9, 9.6] | 5.9% [3.8, 8.1] | 5.7% [3.6, 7.8] |
| More than 500 pesos | 18.5% [15.0, 22.0] | 28.0% [24.0, 32.1] | 32.7% [28.5, 36.9] |
| Total | 100% | 100% | 100% |

**Lower and upper bounds of weighted average WTP in pesos, by scenario**

| | Soda consumption reduction | | + health benefits | | + social benefits | |
|---|---|---|---|---|---|---|
| | Lower | Upper | Lower | Upper | Lower | Upper |
| Assuming maximum WTP of 600 pesos for participants who chose "more than 500 pesos" | 196 | 296 | 238 | 338 | 267 | 368 |
| Assuming maximum WTP 1000 pesos for participants who chose "more than 500 pesos" | 196 | 368 | 238 | 450 | 267 | 500 |

Confidence intervals in brackets.

adds health benefits and 302.6 for the scenario that add social benefits. For those with no intention to reduce soda consumption the average WTP was 162.5, 212.0 and 264.5, respectively.

For the distribution of the WTP intervals, in the soda consumption reduction scenario, 29% are in the lowest WTP interval and 21% at the highest. The proportion of participants in the most upper range increases to 28% in the scenario that provides information on health benefits and to 33% in the scenario that also offers social benefits. We found a higher proportion of those who would like to reduce consumption in the most upper WTP interval compared with those who had no intention to reduce soda consumption (results not shown). The weighted average WTP ranges from 196 to 500 pesos depending on the scenario and the assumptions made on the point estimate value for the open-ended option: when participants chose the option of 500 pesos or more, we estimated the weighted average WTP under two upper bounds: up to 600 and up to 1,000 pesos.

Table 3 shows results from the interval regression estimations. In the soda consumption reduction scenario, the base average WTP was 121.3 pesos, which represents the WTP among non-obese younger men, who are not interested in reducing their consumption but in the group with the lowest expenditure on soda, and in the lowest income group. Those who would like to reduce soda consumption are willing to pay 85.1 pesos more than those who do not (60% more than the base WTP). Participants in the fourth quartile of soda expenditures are willing to pay 66.7 pesos more compared with those in the lower quartile (47% more than the base WTP). Those in the middle and high tertiles of income reported a higher WTP compared to the lowest income group. Results for the second and third scenarios are consistent with those in the first, with higher WTP as additional benefits of reducing soda consumption are made explicit, statistically significant for willingness to reduce intake and household income.

Fig 1 shows the mean predicted values for each variable category from the interval regression. By age group, we found that individuals aged less than 65 years-old showed an average higher WTP compared to older participants—5%, 12%, and 18% in the first, second and third scenarios, respectively. Average WTP increases with income, with substantial differences among income tertiles, most notably in the case of the lowest income group, with respect to the other two. Participants who wanted to reduce soda consumption reported 27%, 25%, and

**Table 3. Interval regressions for the willingness to pay for an intervention that reduces soda consumption, by scenario.**

| Variable | | Soda consumption reduction | + health benefits | + social benefits |
|---|---|---|---|---|
| | | Coefficients from interval regression, 95% confidence interval | | |
| Sex (female = 1) | | 9.3 [-33.8, 52.4] | 12.0 [-42.1, 66.2] | 18.2 [-35.8, 72.7] |
| Age group (>65 years old) | | -10.7 [-53.5, 32.1] | -24.9 [-77.2, 27.3] | -36.2 [-88.7, 16.4] |
| Education (university or higher) | | 22.5 [-141, 59.2] | 42.1+ [-3.9, 88.0] | 48.0* [1.5, 94.4] |
| Household income | | | | |
| | Middle | 52.1** [12.0, 92.3] | 75.7 ** [26.6, 124.7] | 77.0** [25.4, 128.6] |
| | High | 80.2** [35.3, 125.0] | 126.7** [68.5, 184.5] | 138.8** [78.4, 199.2] |
| High soda consumption (> $30 pesos) | | 66.7** [25.8, 107.5] | 100.8** [50.3, 151.1] | 99.1** [46.3, 151.7] |
| Would like to reduce soda consumption | | 85.1** [38.7, 131.5] | 83.4** [24.2, 142.4] | 56.7+ [-3.9, 117.4] |
| Overweight and obese | | 2.1 [-34.7, 39.0] | 10.3 [-37.3, 58.0] | 5.3 [-42.9, 53.4] |
| Constant | | 121.3** [53.4, 189.2] | 115.1* [36.4,193.7] | 170.3** [86.9, 253.7] |

Confidence intervals in brackets. Significance

+ at 10%

* at 5%

** at 1%. Bootstrap robust standard errors, clustered at the household level.

Dependent variable: Willingness to pay.

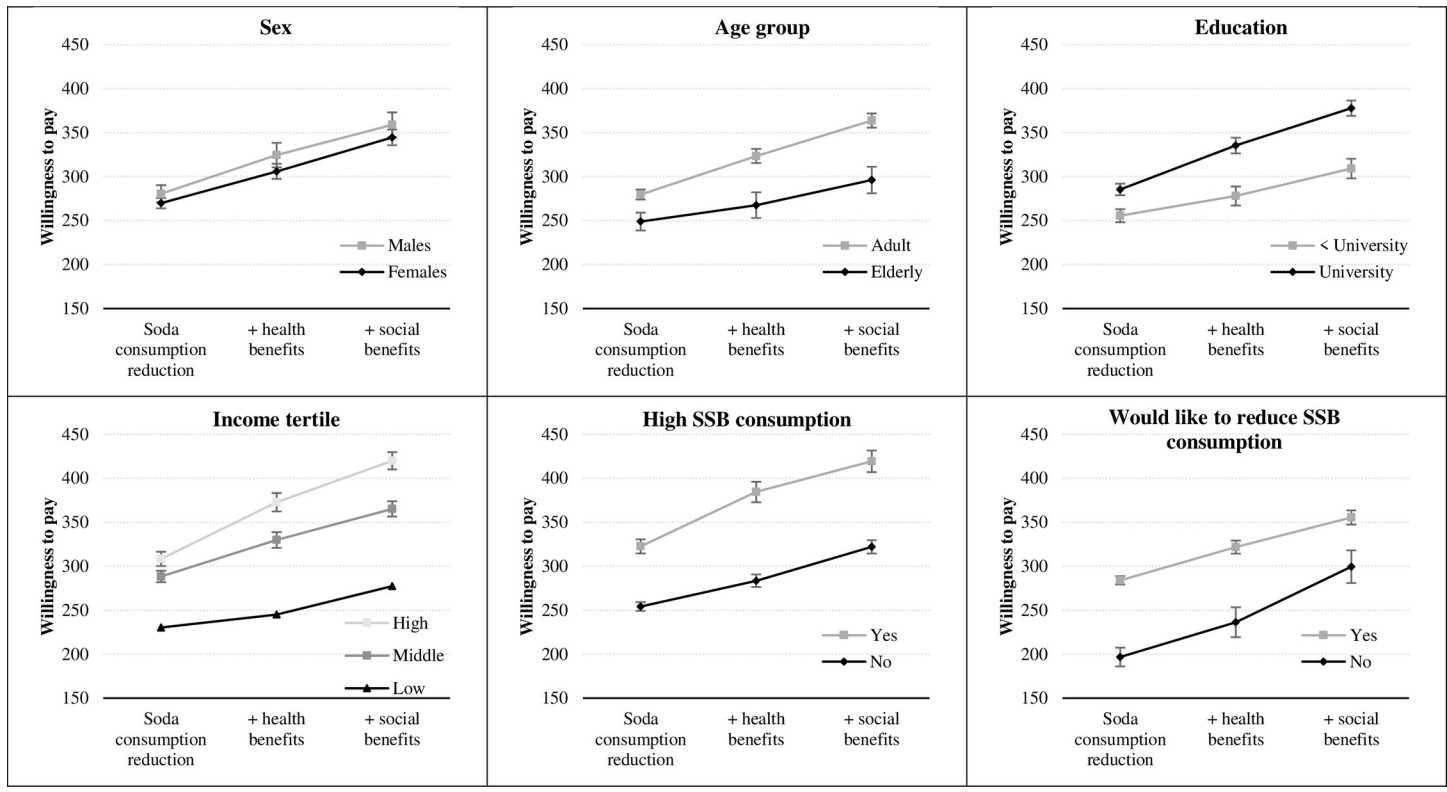

**Fig 1. Average willingness to pay for an intervention that reduces soda consumption, by scenario and variable.**

17% higher WTP compared to those not willing to reduce intake, in the first, second and third scenarios, respectively. Participants with higher soda consumption reported about 20% higher WTP compared to those with lower soda consumption. Overall, the mean adjusted WTP increased from 272.5 pesos in the basic scenario to 310.5 pesos in the second scenario and 348.1 pesos when social benefits are explicit (results not shown).

When we take the ratio of the WTP relative to income, we found that individuals from the lowest income group are willing to pay a more substantial proportion of their income, compared to those in the highest income group (Table 4). For instance, for the low-income group, the WTP for the soda consumption reduction scenario represents 5% of their income compared to 2% and 1% from the middle- and high-income groups, respectively. The results are very similar for the other two scenarios.

We found no statistical differences between the analytical sample (471) and the excluded observations in the covariates of interest (39). Also, we did not find statistically significant differences in age, gender or education between the analytical sample of individuals that participated in the WTP study with participants in the third wave of the HWSC.

**Table 4. Ratio of WTP relative to income, by household income group and WTP scenario.**

| Income group | Soda consumption reduction | + health benefits | + social benefits |
|---|---|---|---|
| Low | 0.050 | 0.052 | 0.059 |
| Middle | 0.019 | 0.021 | 0.023 |
| High | 0.010 | 0.012 | 0.013 |

## Discussion

We estimated the WTP for an intervention that reduces soda consumption by half in a sub-sample of 471 participants in the HWCS. The study provides an estimate of the social value for such an intervention, and therefore of the social value reducing soda consumption in this cohort. Our results showed that individuals with high soda consumption, with the intention to reduce soda consumption, and in the higher household income group, reported higher WTP. We also found that as we included additional benefits in the description of the intervention the respondents increased their WTP (average adjusted WTP increases from 272.5 to 348.4 pesos). Finally, we found that the WTP, as a proportion of income, was higher among the lowest income level group. The results show the potential acceptability of public policies to reduce consumption of perceived harmful products.

Our study also revealed that 87% of respondents reported they would like to reduce SSB consumption. From a welfare perspective, an SSB tax increase would potentially represent a net benefit for those willing to reduce SSB consumption. This result reveals a relatively high awareness of the adverse health effects of soda consumption or at least, the potential value of reducing consumption—although we were not able to measure whether social desirability bias influenced our respondents.

Our findings are consistent with previous literature. A study in the US showed that people were willing to pay about 46 USD for a 50% reduction in childhood obesity policies, which was greater than current per capita public health spending [17]. Although their questions are different from our study, both studies show support from taxpayers to invest in public policies to reduce the burden of chronic diseases.

As in our study, in general, high income individuals declared higher willingness to pay [18]. A study shows that high sugar-sweetened beverage consumers are more responsive to increases in prices [19] which could explain our findings that high soda spending was associated with a greater WTP. In general, for high consumers of harmful goods, a reduction in consumption represents a benefit because it aligns their observed consumption with their rational consumption [20, 21].

The results of the study reveal individual preferences associated with their willingness to reduce soda consumption. The higher WTP is for the scenario that adds social benefits, can be used to frame a policy aimed at reducing soda consumption by including that the money collected from the program could be used to fund treatment of chronic diseases and to provide potable in the most deprived regions and public schools in Mexico.

The study has some limitations. We set-up the upper amount to 500 pesos based on national data on household expenditures from the 2014 National Income and Expenditure Survey [22]. Given that more than 20% of the participants WTP is in the highest interval, we acknowledge the possibility that the real average WTP may be higher had we allowed for a higher upper amount. We acknowledge that imputation using linear regression for the 16 missing values for income could reduce the variability. However, missing values represented only 3.4% of the sample and the literature recognizes that less than 10% of missing values leads to unbiased results [23–25].

We also acknowledge that the representativeness of the results is determined by the HWSC, which includes individuals with an average higher education and income compared to the general population. However, future studies using a representative sample would get a more accurate estimate of the average national WTP for interventions to reduce soda consumption. We did not make specific the duration of the intervention which is a limitation for future studies, nor the specific mechanism of payment, as is recommended elsewhere [17].

The sample size was limited to available funding, we followed the sample used by Cawley in a study that estimated the willingness to pay to reduce childhood obesity (n = 477) [17]. To address the potential limitation of a small sample size, we ran the interval regressions using bootstrap to get robust standard errors.

The National Oceanic and Atmospheric Administration (NOAA) issued some recommendations to maximize the reliability of contingent valuation estimates [20, 26]. Our study complies with most of the recommendations. We applied the questionnaire in a face-to-face or telephone interviews, not on the mail. We also pretested the questionnaire although we failed to include the "would not answer" option. We broke down the WTP by a set of characteristics of interest such as sex, income level, and attitudes towards soda consumption, and body mass index. We also reminded the respondents of their actual budget constraint when considering their willingness to pay as an out-of-pocket expenditure, as the NOAA recommended. We measured "willingness to pay" rather than "willingness to accept," as recommended. We used a subsample of the HWSC, which does not include a probabilistic sample design. However, the WTP subsample is representative of the entire cohort.

The WTP reflects the preference to reduce soda consumption. We acknowledge that we cannot translate this social value as if we would implement an intervention that restricts the choice set. For those who had the intention to reduce soda consumption, the nutrition program aligns their preferences to their habits. For those who did not have the intention to reduce soda consumption, the program changes their preferences. In both cases, the program does not restrict the choice set.

We acknowledge a potential reverse causality if willingness to reduce soda consumption, that we use as an independent variable, causes WTP. We followed two approaches. We first ran the interval regression excluding willingness to reduce soda consumption and found that the results are very similar. We also ran a probit model with a binary variable of willingness to reduce soda consumption as the dependent variable, willingness to pay as independent variable, and the other covariates but the coefficient for WTP was not statistically significant.

In conclusion, our study provides an estimate of how much people value an intervention that helps them reducing soda consumption. Even for individuals reporting they would not like their intake when informed about the positive effects of reducing SSB consumption, their WTP increased, which implies that information on the harmful effects of SSB intake is needed to make better decisions. Our findings illustrate two relevant policy implications. First, interventions aimed at reducing SSB consumption -such as fiscal policies, front of pack labeling or school restrictions on unhealthy food and beverages sales-, represent a benefit for people who are willing to reduce soda consumption (a high percentage in our study). Secondly, the average WTP per person ranged from 272.5 to 348.4 pesos per month. This amount represents the social value on an intervention to reduce soda consumption which provides a benchmark for policy makers to allocating more resources o on obesity prevention strategies and on providing water fountains in the poorest regions on the country and in public schools. Moreover, based on the 2018 National Income and Expenditure Survey, SSB consumers spend $261 monthly pesos (excluding away from home purchases), which is in the range of the WTP. This value the how much SSB consumers value the potential benefits on health from an intervention to reduce SSB consumption.

## Supporting information

**S1 Appendix. Survey instrument (translated from Spanish to English).**
(DOCX)

**S1 Fig. Example of responses in the bidding process for a sequence of questions when the random start value is 100 pesos.**
(DOCX)

## Acknowledgments

We would like to acknowledge the team of interviewers from the Health Workers Cohort Study: Daniela Antunez and Griselda Díaz, as well as Julio César Ruiz for programming the questionnaire in access.

## Author Contributions

**Conceptualization:** M. A. Colchero, Tonatiuh Barrientos-Gutiérrez T., Jorge Salmerón J., Sergio Bautista-Arredondo.

**Data curation:** Carlos M. Guerrero-López C. M.

**Formal analysis:** M. A. Colchero, Carlos M. Guerrero-López C. M., Sergio Bautista-Arredondo.

**Investigation:** M. A. Colchero, Tonatiuh Barrientos-Gutiérrez T., Jorge Salmerón J.

**Methodology:** M. A. Colchero, Carlos M. Guerrero-López C. M., Tonatiuh Barrientos-Gutiérrez T., Sergio Bautista-Arredondo.

**Supervision:** M. A. Colchero.

**Writing – original draft:** M. A. Colchero.

**Writing – review & editing:** M. A. Colchero, Carlos M. Guerrero-López C. M., Tonatiuh Barrientos-Gutiérrez T., Jorge Salmerón J., Sergio Bautista-Arredondo.

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
