## [Decision Letter · Decision Letter 0]

10 Mar 2021

PONE-D-20-34196

Willingness to pay for an intervention that reduces soda consumption among a sample of middle-class adult Mexicans

PLOS ONE

Dear Dr. Colchero,

Thank you for submitting your manuscript to PLOS ONE. After careful consideration, we feel that it has merit but does not fully meet PLOS ONE’s publication criteria as it currently stands. Therefore, we invite you to submit a revised version of the manuscript that addresses the points raised during the review process.

Both reviewers raised substanial concerns and their recommendations are split. One reviewer recommended rejection. Please try to address their concerns as much as you can. I also have a question as I read your paper: the elicited WTP is for reduicng half the soda consumption. how much is half? Can you also estimate the WTP for reducing one liter? I think this number is more relevant for policy makers.

We look forward to receiving your revised manuscript.

Kind regards,

Shihe Fu, Ph.D.

Academic Editor

PLOS ONE

Journal Requirements:

2. Please include additional information regarding the survey or questionnaire used in the study and ensure that you have provided sufficient details that others could replicate the analyses. For instance, if you developed a questionnaire as part of this study and it is not under a copyright license more restrictive than CC-BY, please include a copy, in both the original language and English, as Supporting Information"

Reviewers' comments:

Reviewer's Responses to Questions

**Comments to the Author**

1. Is the manuscript technically sound, and do the data support the conclusions?

Reviewer #1: Yes

Reviewer #2: Partly

2. Has the statistical analysis been performed appropriately and rigorously? 

Reviewer #1: N/A

Reviewer #2: Yes

3. Have the authors made all data underlying the findings in their manuscript fully available?

Reviewer #1: Yes

Reviewer #2: Yes

4. Is the manuscript presented in an intelligible fashion and written in standard English?

Reviewer #1: Yes

Reviewer #2: Yes

5. Review Comments to the Author

Reviewer #1: Mexico has adopted a SSB tax to control the obesity and related chronical diseases. It has been discovered to be significantly decreases the soda consumption. This article contributes to the literature in assessing the willingness to reduce soda consumption and the willingness to pay (WTP) for various interventions that reduces soda consumption. This article focuses on providing knowledges of the benefits and welfare implications of these interventions, including reductions in consumption, health benefits such as reductions in obesity, diabetes and other chronic disease, and using money collected by SSB sales to treatment of chronic diseases and to potable water in the most deprived regions and public schools in Mexico. The author(s) found that people would like to reduce soda consumption, and as more beneficial information is provided, people’s WTP increases.

In general, the topic is interesting. This article fills the gap in exploring the relation between WTP for soda drinks and interventions—health benefits and social benefits.

Major comments

1.There are 1241 obs. in the third wave of HWSC, however the analytical sample is 470 in this study. Why did the rest of the individuals not participate in the WTP study? Is there any sample selection criterion? Are sample weights used in the estimation?

2.In page 8, “We also assessed whether the subsample of the WTP study was representative of the HWSC by testing for statistical differences in the co-variables included in the analysis between our analytical sample and all participants of the cohort in the third wave.” Can you provide the details? Whether the difference is significant or not for each explanatory variable?

3.Although by reading the whole manuscript, we are able to figure out that interval regression is applied in the empirical estimation. I strongly suggest the author(s) to provide detailed description on the methodology used in the manuscript.

4.Sample size is too small (471). Robustness check should be conducted. Proper empirical approach should be used to address the data limitation, i.e., bootstrap.

5.Whether the estimation (Table 3) is clustered by household or else, there is no specification on it.

6.I think authors should focus more on potential policy implications regarding to various responses to tax revenue recycling. For example, if individuals are discovered to be more altruistic –using the collected money to help the poor, what kind of policy should the government adopt? Or else, individuals are revealed to be more conscious of heath benefits, what kind of scenario should be used?

Reviewer #2: In this paper, the authors assess the willingness to pay (WTP) for an intervention that would reduces soda consumption by half in Mexico. Using contingent valuation methods in a sample of 471 Mexican adults from a cohort study, they find that high soda consumption, willingness to reduce soda consumption , higher household income, and additional information about the health and social benefits of reducing soda consumption are associated with higher WTP. They also find that the WTP, as a proportion of income, is higher for the lowest income level.

This paper contributes to the policy debate on the social value of a reduction in soda consumption in Mexico by directly eliciting the population's WTP to reduce soda consumption. However, the contribution is severely compromised by the way the intervention was interpreted in the questionnaire.

Question 2 in the survey asks ``Imagine there is a nutrition program that aims to change your soda consumption habits that would reduce in half your current consumption as long you remain in the program. This program does not require you to take pills or any surgery. You are supposed to pay for it, and it is not provided by the government or any other Health institution." The wording implies that the method through which the hypothetical intervention would reduce soda consumption is a change in preferences without restricting the choice set, which differs from what actual interventions usually do, such as a tax on soda consumption and regulation of the beverage in schools. In the latter cases, there might be large reduction in consumer's maximized utility caused by the shrinking of the choice set. Because they are not informed about these potential losses, the respondents will considerably overestimate their WTP.

Minor Comments:

[1] It is not clear if sample weights are used to infer the WTP of the population. Weighting could be important when the demographic composition in the sample is different from that in the population to the extent that 75% of the sample are female. \\\\

[2] One of the independent variables in the regression analysis is the 0-1 variable ``would like to reduce soda." It would be a bad control if it is an outcome rather than a cause of the WTP to reduce soda.

6. PLOS authors have the option to publish the peer review history of their article (what does this mean?). If published, this will include your full peer review and any attached files.

Reviewer #1: No

Reviewer #2: No

---

## [Author Response · Author response to Decision Letter 0]

11 May 2021

Reviewers' comments:

Reviewer #1: Mexico has adopted a SSB tax to control the obesity and related chronical diseases. It has been discovered to be significantly decreases the soda consumption. This article contributes to the literature in assessing the willingness to reduce soda consumption and the willingness to pay (WTP) for various interventions that reduces soda consumption. This article focuses on providing knowledges of the benefits and welfare implications of these interventions, including reductions in consumption, health benefits such as reductions in obesity, diabetes and other chronic disease, and using money collected by SSB sales to treatment of chronic diseases and to potable water in the most deprived regions and public schools in Mexico. The author(s) found that people would like to reduce soda consumption, and as more beneficial information is provided, people’s WTP increases.

In general, the topic is interesting. This article fills the gap in exploring the relation between WTP for soda drinks and interventions—health benefits and social benefits.

Major comments

1.There are 1241 obs. in the third wave of HWSC, however the analytical sample is 470 in this study. Why did the rest of the individuals not participate in the WTP study? Is there any sample selection criterion? Are sample weights used in the estimation?

Response: The sample size was limited by the available funding; this was a sub study funded independently of the cohort. There are few studies that could be used to estimate sample size. We followed the sample used by Cawley in a study that estimated the willingness to pay to reduce childhood obesity (n=477). We did not use weights for the estimation. We added the following in the discussion section.

“The sample size was limited to available funding, we followed the sample used by Cawley in a study that estimated the willingness to pay to reduce childhood obesity (n=477). To address the potential limitation of a small sample size, we ran the interval regressions using bootstrap to get robust standard errors.“ 

2.In page 8, “We also assessed whether the subsample of the WTP study was representative of the HWSC by testing for statistical differences in the co-variables included in the analysis between our analytical sample and all participants of the cohort in the third wave.” Can you provide the details? Whether the difference is significant or not for each explanatory variable?

Response: We appreciate the comment. We added the following:

Methods

“We also assessed whether the subsample of the WTP study was representative of the HWSC by testing for statistical differences in age, gender and education between our analytical sample and all participants of the cohort in the third wave”. 

Results

“We did not find statistically significant differences in age, gender or education between the analytical sample of individuals that participated in the WTP study with participants in the third wave of the HWSC”. 

3.Although by reading the whole manuscript, we are able to figure out that interval regression is applied in the empirical estimation. I strongly suggest the author(s) to provide detailed description on the methodology used in the manuscript.

Response: In the new version of the manuscript, we provide more details on the methods used for empirical estimation. As mentioned in the methods (statistical analysis), we first describe the distribution of the WTP intervals for the three scenarios then we applied interval regression for empirical estimation. We added the following to provide more details:

We first describe the unadjusted distribution of the WTP for the three scenarios…

For empirical estimation, we ran an interval regression for each WTP scenario. Interval regressions are used when the dependent variable is expressed as a continuous variable or by intervals, with left- or right-censored data….

The independent variables included in the interval regression model were selected based on two criteria….

4.Sample size is too small (471). Robustness check should be conducted. Proper empirical approach should be used to address the data limitation, i.e., bootstrap.

Response: We thank the reviewer for the comment. We conducted a robustness check using bootstrap and the results are almost identical. We left the specification with robust standard errors as the main specification. We added the following in the methods and results section:

“We used bootstrap with 1000 replications and variance-covariance estimation to get bootstrap standard errors”.

5.Whether the estimation (Table 3) is clustered by household or else, there is no specification on it.

Response: We thank the reviewer for the comment. We have about 30% of participants in the same household. The new version of the models includes clustering at the household level. We added the following: 

“About 30% of participants belong to the same household so we included clustering at the household level in the interval regression”. 

6.I think authors should focus more on potential policy implications regarding to various responses to tax revenue recycling. For example, if individuals are discovered to be more altruistic –using the collected money to help the poor, what kind of policy should the government adopt? Or else, individuals are revealed to be more conscious of heath benefits, what kind of scenario should be used?

Response: We agree with the reviewer. The results or the study reveal their preferences and how to frame a policy. We added the following:

“The results of the study reveal individual preferences associated with their willingness to reduce soda consumption. The higher WTP is for the scenario that adds social benefits, can be used to frame a policy aimed at reducing soda consumption by including that the money collected from the program could be used to fund treatment of chronic diseases and to provide potable in the most deprived regions and public schools in Mexico.”

Reviewer #2: In this paper, the authors assess the willingness to pay (WTP) for an intervention that would reduces soda consumption by half in Mexico. Using contingent valuation methods in a sample of 471 Mexican adults from a cohort study, they find that high soda consumption, willingness to reduce soda consumption , higher household income, and additional information about the health and social benefits of reducing soda consumption are associated with higher WTP. They also find that the WTP, as a proportion of income, is higher for the lowest income level.

This paper contributes to the policy debate on the social value of a reduction in soda consumption in Mexico by directly eliciting the population's WTP to reduce soda consumption. However, the contribution is severely compromised by the way the intervention was interpreted in the questionnaire.

Question 2 in the survey asks ``Imagine there is a nutrition program that aims to change your soda consumption habits that would reduce in half your current consumption as long you remain in the program. This program does not require you to take pills or any surgery. You are supposed to pay for it, and it is not provided by the government or any other Health institution." The wording implies that the method through which the hypothetical intervention would reduce soda consumption is a change in preferences without restricting the choice set, which differs from what actual interventions usually do, such as a tax on soda consumption and regulation of the beverage in schools. In the latter cases, there might be large reduction in consumer's maximized utility caused by the shrinking of the choice set. Because they are not informed about these potential losses, the respondents will considerably overestimate their WTP.

Response: We agree with the reviewer. We added the following in the discussion section.

“The WTP reflects the preference to quit or to reduce soda consumption. We acknowledge that we cannot translate this social value as if we would implement an intervention that restricts the choice set.” 

Minor Comments:

[1] It is not clear if sample weights are used to infer the WTP of the population. Weighting could be important when the demographic composition in the sample is different from that in the population to the extent that 75% of the sample are female. \\\\

Response: There are no sample weights in the data set. As acknowledged in the discussion section, the cohort is not representative of the population as it is for the Health Workers Cohort Study. However, we tested for statistical differences in basic characteristics between our analytic sample and the third wave of the Health Workers Cohort Study and we did not find significant differences. 

[2] One of the independent variables in the regression analysis is the 0-1 variable ``would like to reduce soda." It would be a bad control if it is an outcome rather than a cause of the WTP to reduce soda.

Response: We understand the concern. We ran the model without the variable and the results are very similar. We also ran a probit model with the “would like to reduce soda” as the dependent variable and WTP as an independent variable, adjusting for the same variables and we did not find a significant association. We added the following in the discussion section:

“We acknowledge a potential reverse causality if willingness to reduce soda consumption, that we use as an independent variable, causes WTP. We followed two approaches. We first ran the interval regression excluding willingness to reduce soda consumption and found that the results are very similar. We also ran a probit model with a binary variable of willingness to reduce soda consumption as the dependent variable, willingness to pay as independent variable, and the other covariates but the coefficient for WTP was not statistically significant”.

---

## [Decision Letter · Decision Letter 1]

16 Jun 2021

PONE-D-20-34196R1

Willingness to pay for an intervention that reduces soda consumption among a sample of middle-class adult Mexicans

PLOS ONE

Dear Dr. Colchero,

Thank you for submitting your manuscript to PLOS ONE. After careful consideration, we feel that it has merit but does not fully meet PLOS ONE’s publication criteria as it currently stands. Therefore, we invite you to submit a revised version of the manuscript that addresses the points raised during the review process.

Your paper is close to be publishable. One reviewer recommended accaptance and another provided a few useful comments for a minor revision.

We look forward to receiving your revised manuscript.

Kind regards,

Shihe Fu, Ph.D.

Academic Editor

PLOS ONE

Journal Requirements:

Reviewers' comments:

Reviewer's Responses to Questions

**Comments to the Author**

1. If the authors have adequately addressed your comments raised in a previous round of review and you feel that this manuscript is now acceptable for publication, you may indicate that here to bypass the “Comments to the Author” section, enter your conflict of interest statement in the “Confidential to Editor” section, and submit your "Accept" recommendation.

Reviewer #1: All comments have been addressed

Reviewer #2: (No Response)

2. Is the manuscript technically sound, and do the data support the conclusions?

Reviewer #1: Yes

Reviewer #2: Partly

3. Has the statistical analysis been performed appropriately and rigorously? 

Reviewer #1: Yes

Reviewer #2: N/A

4. Have the authors made all data underlying the findings in their manuscript fully available?

Reviewer #1: Yes

Reviewer #2: No

5. Is the manuscript presented in an intelligible fashion and written in standard English?

Reviewer #1: Yes

Reviewer #2: Yes

6. Review Comments to the Author

Reviewer #1: The author has addressed most of my questions. Although sample size and its representativeness still bothers, the use of bootstrap method and appropriate empirical approaches could to some extent address it.

Reviewer #2: Question 2 in the survey asks ``Imagine there is a nutrition program that aims to change your soda consumption habits that would reduce in half your current consumption as long you remain in the program. This program does not require you to take pills or any surgery. You are supposed to pay for it, and it is not provided by the government or any other Health institution." The wording implies that the method through which the hypothetical intervention would reduce soda consumption is a change in preferences without restricting the choice set, which differs from what actual interventions usually do, such as a tax on soda consumption and regulation of the beverage in schools. In the latter cases, there might be large reduction in consumer's maximized utility caused by the shrinking of the choice set. Because they are not informed about these potential losses, the respondents will considerably overestimate their WTP.

Response: We agree with the reviewer. We added the following in the discussion section.

“The WTP reflects the preference to quit or to reduce soda consumption. We acknowledge that we cannot translate this social value as if we would implement an intervention that restricts the choice set.”

Response to authors’ response: The word “quit” plays a key role in clarifying that the WTP is for removing soda addiction instead of restricting soda consumption. The ambiguity between the two was central in the issue in my previous comments. It might be helpful to solve the issue if you could also implement this clarification to the ambiguous “willingness to reduces SSB consumption” in the abstract and many other places in the manuscript.

Minor Comments:

[1] It is not clear if sample weights are used to infer the WTP of the population. Weighting could be important when the demographic composition in the sample is different from that in the population to the extent that 75% of the sample are female.

Response: There are no sample weights in the data set. As acknowledged in the discussion section, the cohort is not representative of the population as it is for the Health Workers Cohort Study. However, we tested for statistical differences in basic characteristics between our analytic sample and the third wave of the Health Workers Cohort Study and we did not find significant differences.

Response to authors’ response: I have no further issue with sample weights.

[2] One of the independent variables in the regression analysis is the 0-1 variable ``would like to reduce soda." It would be a bad control if it is an outcome rather than a cause of the WTP to reduce soda.

Response: We understand the concern. We ran the model without the variable and the results are very similar. We also ran a probit model with the “would like to reduce soda” as the dependent variable and WTP as an independent variable, adjusting for the same variables and we did not find a significant association. We added the following in the discussion section:

“We acknowledge a potential reverse causality if willingness to reduce soda consumption, that we use as an independent variable, causes WTP. We followed two approaches. We first ran the interval regression excluding willingness to reduce soda consumption and found that the results are very similar. We also ran a probit model with a binary variable of willingness to reduce soda consumption as the dependent variable, willingness to pay as independent variable, and the other covariates but the coefficient for WTP was not statistically significant”.

Response to authors’ response: Does “Would like to reduce soda=0” imply “WTP=0” in theory? How does isolating the effect of “WTP>0” on WTP help us understand WTP?

7. PLOS authors have the option to publish the peer review history of their article (what does this mean?). If published, this will include your full peer review and any attached files.

Reviewer #1: No

Reviewer #2: No

---

## [Author Response · Author response to Decision Letter 1]

28 Jun 2021

Reviewers' comments:

Reviewer #1: The author has addressed most of my questions. Although sample size and its representativeness still bothers, the use of bootstrap method and appropriate empirical approaches could to some extent address it.

Response: Thank you.

Reviewer #2: Question 2 in the survey asks ``Imagine there is a nutrition program that aims to change your soda consumption habits that would reduce in half your current consumption as long you remain in the program. This program does not require you to take pills or any surgery. You are supposed to pay for it, and it is not provided by the government or any other Health institution." The wording implies that the method through which the hypothetical intervention would reduce soda consumption is a change in preferences without restricting the choice set, which differs from what actual interventions usually do, such as a tax on soda consumption and regulation of the beverage in schools. In the latter cases, there might be large reduction in consumer's maximized utility caused by the shrinking of the choice set. Because they are not informed about these potential losses, the respondents will considerably overestimate their WTP.

Response: We agree with the reviewer. We added the following in the discussion section.

“The WTP reflects the preference to quit or to reduce soda consumption. We acknowledge that we cannot translate this social value as if we would implement an intervention that restricts the choice set.”

Response to authors’ response: The word “quit” plays a key role in clarifying that the WTP is for removing soda addiction instead of restricting soda consumption. The ambiguity between the two was central in the issue in my previous comments. It might be helpful to solve the issue if you could also implement this clarification to the ambiguous “willingness to reduces SSB consumption” in the abstract and many other places in the manuscript.

Response: We agree with the reviewer. We are not measuring willingness to reduce SSB consumption. The questionnaire asks individuals for their intention to reduce soda consumption. We replaced “willingness to reduce soda consumption” to “intention to reduce soda consumption” all over the manuscript.

We also agree that we are not restricting the choice set.

We added the following:

“The WTP reflects the preference to reduce soda consumption. We acknowledge that we cannot translate this social value as if we would implement an intervention that restricts the choice set. For those who had the intention to reduce soda consumption, the nutrition program aligns their preferences to their habits. For those who did not have the intention to reduce soda consumption, the program changes their preferences. In both cases, the program does not restrict the choice set.”

Minor Comments:

[1] It is not clear if sample weights are used to infer the WTP of the population. Weighting could be important when the demographic composition in the sample is different from that in the population to the extent that 75% of the sample are female.

Response: There are no sample weights in the data set. As acknowledged in the discussion section, the cohort is not representative of the population as it is for the Health Workers Cohort Study. However, we tested for statistical differences in basic characteristics between our analytic sample and the third wave of the Health Workers Cohort Study and we did not find significant differences.

Response to authors’ response: I have no further issue with sample weights.

Response: Thank you.

[2] One of the independent variables in the regression analysis is the 0-1 variable ``would like to reduce soda." It would be a bad control if it is an outcome rather than a cause of the WTP to reduce soda.

Response: We understand the concern. We ran the model without the variable and the results are very similar. We also ran a probit model with the “would like to reduce soda” as the dependent variable and WTP as an independent variable, adjusting for the same variables and we did not find a significant association. We added the following in the discussion section:

“We acknowledge a potential reverse causality if willingness to reduce soda consumption, that we use as an independent variable, causes WTP. We followed two approaches. We first ran the interval regression excluding willingness to reduce soda consumption and found that the results are very similar. We also ran a probit model with a binary variable of willingness to reduce soda consumption as the dependent variable, willingness to pay as independent variable, and the other covariates but the coefficient for WTP was not statistically significant”.

Response to authors’ response: Does “Would like to reduce soda=0” imply “WTP=0” in theory? How does isolating the effect of “WTP>0” on WTP help us understand WTP?

Response: We are not assuming that those who did not have an intention to reduce soda had a willingness to pay equal to zero since, as mentioned above, the program can change their preferences. Actually, their WTP is greater than zero, lower than those who had the intention to reduce soda but not zero. We added in table 2, the average WTP by intention to reduce soda consumption by scenario.

We added the following in the methods section:

“We first described the average WTP by intention to reduce soda consumption by scenario and the unadjusted distribution of the WTP for the three scenarios: 1) cut in soda consumption, 2) cut in soda consumption + explicitly accounting for individual health benefits, 3) adding social benefits.”

We added the following in the results section (description of table 2):

“Table 2 shows the average WTP by intention to reduce soda consumption and the distribution of the WTP intervals by scenario. Those who had the intention to reduce soda consumption had a higher WTP compared to those who had not intention to reduce soda consumption but their WTP. The average WTP for the soda consumption reduction scenario was 244.9 pesos for those with the intention to reduce soda consumption, 280.5 for the scenario that adds health benefits and 302.6 for the scenario that add social benefits. For those with no intention to reduce soda consumption the average WTP was 162.5, 212.0 and 264.5, respectively.”

---

## [Editor Report · Decision Letter 2]

12 Jul 2021

Willingness to pay for an intervention that reduces soda consumption among a sample of middle-class adult Mexicans

PONE-D-20-34196R2

Dear Dr. Colchero,

I have read your second-round revsied version and I think you have addressed the referee comments well. We’re pleased to inform you that your manuscript has been judged scientifically suitable for publication and will be formally accepted for publication once it meets all outstanding technical requirements.

Kind regards,

Shihe Fu, Ph.D.

Academic Editor

PLOS ONE
---

## [Editor Report · Acceptance letter]

21 Jul 2021

PONE-D-20-34196R2 

Willingness to pay for an intervention that reduces soda consumption among a sample of middle-class adult Mexicans 

Dear Dr. Colchero:

I'm pleased to inform you that your manuscript has been deemed suitable for publication in PLOS ONE. Congratulations! Your manuscript is now with our production department. 

Kind regards, 

on behalf of

Dr. Shihe Fu 

Academic Editor

PLOS ONE